# Combining Ability and Inheritance Nature of Agronomic Traits and Resistance to Pink Stem (*Sesamia cretica*) and Purple-Lined (*Chilo agamemnon*) Borers in Maize

**DOI:** 10.3390/plants12051105

**Published:** 2023-03-01

**Authors:** Saad N. AL-Kahtani, Mohamed M. Kamara, El-Kazafy A. Taha, Nabil El-Wakeil, Ahmed Aljabr, Kareem M. Mousa

**Affiliations:** 1Arid Land Agriculture Department, College of Agricultural Sciences & Foods, King Faisal University, P.O. Box 400, Al-Ahsa 31982, Saudi Arabia; 2Department of Agronomy, Faculty of Agriculture, Kafrelsheikh University, Kafr El-Sheikh 33516, Egypt; 3Economic Entomology Department, Faculty of Agriculture, Kafrelsheikh University, Kafr El-Sheikh 33516, Egypt

**Keywords:** *Zea mays*, borers resistance, high-yielding, combining ability, insect infestation

## Abstract

The pink stem borer (PSB), *Sesamia cretica* (Lepidoptera: Noctuidae) purple-lined borer (PLB), *Chilo agamemnon* (Lepidoptera: Crambidae) and European corn borer *Ostrinia nubilalis*, (Lepidoptera: Crambidae) are considered the most devastating insect pests of maize production in the Mediterranean region. The frequent use of chemical insecticides has resulted in the evolution of resistance to various insect pests as well as the pernicious impact on natural enemies and environmental hazardousness. Therefore, developing resistant and high-yielding hybrids is the best economic and environmental approach to cope with these destructive insects. Accordingly, the objective of the study was to estimate the combining ability of maize inbred lines (ILs), identify promising hybrids, determine gene action controlling agronomic traits and resistance to PSB and PLB, and investigate inter-relationships among evaluated traits. A half-diallel mating design was employed to cross seven diverse maize inbreds to generate 21 F_1_ hybrids. The developed F_1_ hybrids, alongside high-yielding commercial check hybrid (SC-132), were assessed in field trials for two years under natural infestation. Substantial variations were obtained among the evaluated hybrids for all recorded characteristics. The non-additive gene action was major for grain yield and its contributing traits, while the additive gene action was more important in controlling the inheritance of PSB and PLB resistance. The inbred line IL1 was identified to be a good combiner for earliness and developing short-stature genotypes. Additionally, IL6 and IL7 were recognized as excellent combiners to enhance resistance to PSB, PLB and grain yield. The hybrid combinations IL1×IL6, IL3×IL6, and IL3×IL7 were determined to be excellent specific combiners for resistance to PSB, PLB and grain yield. Strong positive associations were identified among grain yield, its related traits, and resistance to PSB and PLB. This implies their importance as useful traits for indirect selection for improving grain yield. Otherwise, the resistance against PSB and PLB was negatively associated with the silking date, indicating that earliness would be favorable for escaping from the borer’s attack. It could be concluded that the inheritance of PSB and PLB resistance can be governed by the additive gene effects, and the IL1×IL6, IL3×IL6, and IL3×IL7 hybrid combinations can be recommended as excellent combiners for resistance to PSB and PLB and good yield.

## 1. Introduction

Maize is the third most essential cereal crop in terms of grown area and total production [1,2]. Globally, it is broadly used for edible oil, animal feed, food, and fuel [3,4,5]. It is considered a vital source of national income for many countries [6,7]. The cultivated area of maize exceeds 202 million hectares which produces nearly 1.1 billion tons of grains, and this production is expected to attain 1.4 billion tons by the year 2030 [1]. In Egypt, maize production is scanty to cope with the challenges of a fast-growing population [8,9]. This gap could be reduced by improving the productivity of the commercial hybrids and, accordingly, the total yield production [10]. Otherwise, maize productivity is restricted by several biotic and abiotic stresses. Various insect pests are infesting maize during the cultivation period causing a considerable loss in crop production, particularly in temperate areas and Mediterranean countries. The pink stem borer (PSB), *Sesamia cretica* Led. (Fam. Noctuidae) and the purple-lined borer (PLB), *Chilo agamemnon* Bles. (Fam. Crambidae) are lepidopteran corn borers that attack maize fields throughout different plant growth stages [11,12]. In Mediterranean countries, *S. cretica* attacks newly emerged maize plants that are less than five weeks old in June and July. Its females lay eggs under the leaf sheath, then the larvae enter the stem and feed deep inside the whorl, causing the death of the feeding point. In contrast, *C. agamemnon* invades older plants and may girdle them causing breakage of the plant’s stem [13]. Feeding on the whorl leaves destroys the young plants and causes dead hearts. In addition, the tunnels are caused by larvae entering the stem, weakening the plant and consequently increasing stalk lodging and reducing grain yield. A yield loss between 20 and 70%, depending on the severity, can be incurred in the infested fields [14]. The frequent use of synthesized chemical insecticides to control herbivore insect pests has developed resistance to many conventional chemical insecticides [15]. In addition to the pernicious impact on natural enemies and the environmental hazardousness [16]. In this regard, the integration and combination of different approaches, such as developing resistant hybrids, exploiting entomopathogenic microorganisms, and applying natural materials and plant extracts as insecticides, are favorable for suppressing insect pest outbreaks [17,18]. However, breeding hybrids for resistance to PSB and PLB with desirable agronomic traits seem to be the cheapest and safest approach to controlling techniques to reduce yield loss for smallholder farmers. Several attempts are in progress to develop borer-resistant and high-yielding hybrids, particularly under the current climate change. Despite these efforts achieving high grain yield combined with high resistance remains restricted [19].

Developing newly resistant and high-yielding hybrids depends mainly on the identification of used parents [20,21,22]. The diallel mating scheme has been broadly used in maize breeding programs to explore the general combining ability (GCA), and specific combining ability (SCA) effects and identify the best combiners. Furthermore, it determines the inheritance nature, which is accountable for the expression of important traits [23,24,25]. This method is valuable for identifying proper parents and their corresponding crosses for exploiting hybrid vigor in maize improvement [26]. The GCA indicates the presence of genes with additive effects, while the SCA reveals the predominance of genes with non-additive effects. The non-additive gene action was reported to be more valuable than the additive effect in modulating grain yield [27,28,29]. Otherwise, the additive genetic impact was elucidated to be important for most traits of resistance to corn borers [19,30]. Nevertheless, inadequate information is reported regarding the combining ability of PSB and PLB resistance in maize in Mediterranean countries. Exploring the combining ability and inheritance nature is crucial for developing resistant and high-yielding hybrids [31,32]. Therefore, this study aimed at (1) exploring GCA for the assessed inbred lines (ILs) and SCA of their hybrids; (2) depicting the inheritance nature of the evaluated agronomic traits and resistance to PSB and PLB; (3) determining resistant and high-yielding hybrids; and (4) assessing the relationship among evaluated traits.

## 2. Results

### 2.1. Analysis of Variance for Borers Resistance Agronomic Traits

The analysis of variance indicated substantial variation among the evaluated hybrids (Table 1). Furthermore, splitting the hybrid effect into GCA and SCA components indicated that general combining ability (GCA) and specific combining ability (SCA) were significant for all evaluated traits. The interaction effects of GCA×Y and SCA×Y were significant for all the studied traits. The ratio of GCA/SCA was less than the unity for all assessed traits except DTS, EL, and NKPR. Additionally, the magnitude of GCA×Y interaction was greater than that of SCA×Y interaction for all assessed traits except PH and HKW. In comparison to agronomic traits, the ANOVA for the purple-lined borer (PLB) and the pink-stem borer (PSB) resistance exhibited significant differences among the hybrids (Table 1). The variances due to GCA and SCA were noticed to be highly significant, and GCA/SCA ratio was higher than the unity. Additionally, the magnitude of GCA×Y interaction was superior to that of SCA×Y interaction (Table 1).

### 2.2. Performance of the Assessed Hybrids

The mean performance of the 21 F_1_ hybrids and the commercial check hybrid (SC-132) for agronomic traits are shown in Table 2. Days to 50% silking (DTS) varied from 59.17 (IL1×IL4) to 70.17 days (IL5×IL6) with an average of 64.14 days. Four crosses, IL1×IL4, IL1×IL7, IL3×IL6, and IL4×IL7, were substantially earlier than the commercial check hybrid. The average plant height (PH) was 160.73 cm, ranging from 137.67 to 194.50 cm. The hybrid IL4×IL5 had the tallest plant height, while the hybrid IL2×IL3 recorded the shortest plant height. Compared to the commercial hybrid, five hybrids, IL1×IL3, IL1×IL7, IL2×IL3, IL2×IL5, and IL4×IL7, were noticeably shorter than the commercial hybrid. Ear length (EL) varied from 13.25 to 17.18, with an average of 15.0. Three crosses, IL1×IL6, IL2×IL6, and IL4×IL5, exhibited considerably higher values of ear length than the commercial hybrid. The average ear diameter (ED) was 4.41 cm varying from 3.57 to 5.04 cm. Two hybrids, IL1×IL6 and IL2×IL3, substantially exceeded the commercial hybrid. Likewise, the number of rows/ear (NRPE) varied from 11.20 to 15.0, with an average of 13.48. The crosses IL1×IL6, IL2×IL5, IL2×IL7, IL3×IL5, IL3×IL6, IL4×IL5, and IL6×IL7 had considerably superior NRPE than the commercial hybrid. The number of kernels/row (NKPR) was 29.55, varying from 23.83 (IL1×IL6) to 33.35 (IL4×IL5). Four hybrid combinations, IL1×IL6, IL3×IL6, IL4×IL5, and IL6×IL7, demonstrated considerably superior values than the commercial hybrid. In the same context, the heaviest hundred kernel weight (HKW) was obtained by the hybrid IL6×IL7 (34.17 g), whereas IL1×IL7 presented the lowest kernel weight (24.83 g). In addition, the five hybrids, IL1×IL6, IL2×IL7, IL3×IL7, IL4×IL5, and IL6×IL7, revealed superior HKW than the commercial hybrid. Grain yield (GY) diverged from 3.32 to 7.92 ton ha^−1^ with an average of 4.95 ton ha^−1^. Noticeably, four hybrids IL1×IL6, IL3×IL7, IL4×IL5 and IL6×IL7, significantly out-yielded the check hybrid by 14.58%, 18.93%, 29.42% and 22.63%, respectively. The resistance index of the purple-lined borer (PLB) varied from 45 to 96.67%, with an average of 82.94%. Six hybrids showed high resistance (>95%). In addition, ten hybrids were significantly superior for PLB resistance than the check hybrid (Figure 1). The resistance to the pink stem borer (PSB) ranged from 77.50 to 100%, with an average of 91.55%. Nine hybrids showed high resistance (>95%). In addition, the hybrids IL1×IL5, IL1×IL6 IL3×IL6, IL3×IL7, IL4×IL5, and IL6×IL7 demonstrated considerably higher PSB resistance than the check hybrid (Figure 1). Furthermore, interestingly, the crosses IL1×IL6, IL3×IL7, IL4×IL5 and IL6×IL7 and L5×T2 displayed high resistance to PLB and PSB and also high grain yield. These cross combinations could be exploited to improve corn borer resistance and grain yield.

### 2.3. Genotypic Classification Based on Agronomic Traits and Borers Resistance Measurements

Agronomic traits and resistance of PLB and PSB measurements were employed to categorize the assessed hybrids into diverse clusters. Using hierarchical clustering, the evaluated hybrids were divided into four groups based on the agronomic traits (Figure 2a). Group A consisted of four hybrids that possessed the superior agronomic performance; group B comprised three hybrids; group C contained seven hybrids that had intermediate values of agronomic traits; whereas group D comprised seven hybrids that produced the worst agronomic performance. Similarly, the genotypes were classified into four groups according to resistance measurements of corn borers (PLB and PSB) (Figure 2b). Group A contained nine hybrids that exhibited the uppermost values of corn borer resistance. Consequently, these genotypes could be considered resistant hybrids. Groups B and C comprised nine hybrids with intermediate values, while group D contained three hybrids with the lowest values of corn borer resistance (PLB and PSB); hence, they are deemed sensitive hybrids.

### 2.4. General Combining Ability Effects (GCA)

Highly positive and significant GCA effects were considered for all recorded agronomic traits except DTS, PH, and EH, where negative values are preferred. The GCA effects for assessed ILs (Table 3) showed that IL1 and IL4 exhibited the uppermost significant and negative GCA effects for DTS. The desirable combiners for PH and EH were IL1 and IL2, which had negative and desirable GCA effects. On the contrary, the superior positive and significant GCA effects were expressed by IL4 and IL6 for EL; IL7 for ED; IL7 for NRPE; IL3, IL5, and IL6 for NKPR; IL6 and IL7 for HKW and IL6, IL7 for GY. For PLB and PSB resistance, positive GCA effects are preferred. The inbred lines, IL5, IL6, and IL7, demonstrated a substantially positive GCA effect for PLB and PSB.

### 2.5. Specific Combining Ability Effects (SCA)

The SCA effects implied that the favorable significant and negative effects for DTS were assigned for the crosses IL1×IL4, IL1×IL6, IL2×IL5, IL2×IL7, and IL3×IL6 (Table 4). Likewise, significant and negative SCA effects for PH were displayed by IL1×IL7, IL2×IL3, IL2×IL5, IL3×IL6, and IL4×IL7. The hybrids IL1×IL5, IL2×IL3, IL4×IL7 and IL5×IL6 possessed significantly negative SCA effects for EH. In contrast, the largest significant and positive SCA effects for El were recorded by IL1×IL2, IL1×IL6, IL3×IL4, IL3×IL7 and IL4×IL5. Meanwhile, the crosses IL1×IL6 and IL4×IL5 exhibited maximum and positive SCA effects for ED. The desirable SCA effect for NRPE was noticed in the hybrids IL1×IL6, IL3×IL6, IL3×IL7, IL4×IL5 and IL6×IL7. Likewise, the crosses IL1×IL5, IL1×IL6, IL4×IL5, and IL6×IL7 demonstrated a significant and positive SCA effect for NKPR. The uppermost significantly positive SCA values for HKW were shown by the hybrids IL1×IL4, IL1×IL6, IL3×IL4, IL3×IL6, IL4×IL5 and IL6×IL7. The hybrids IL1×IL6, IL2×IL5, IL3×IL7, IL4×IL5 and IL6×IL7 showed significantly positive and desirable SCA effect for GY. The highest positive and negative SCA values for PLB were displayed by the hybrids IL1×IL5, IL1×IL6, IL1×IL7, IL2×IL4, IL2×IL5, IL3×IL6 and IL3×IL7. However, the crosses IL1×IL5, IL1×IL6, IL3×IL6 and IL3×IL exhibited significant and positive SCA effects for PSB. Noticeably, no hybrid concurrently demonstrated desirable SCA effects for all studied traits. On the other hand, certain hybrids showed valuable effects for GY and had advantageous SCA effects for one or more of its related traits. The hybrid IL4×IL5 exhibited favorable SCA effects for EL, NRPE, HKW, NKPR and GY, and the cross IL6×IL7 displayed advantageous SCA effects for NRPE, HKW, NKPR and GY.

### 2.6. Interrelationship among Measured Traits

The principal components (PCA) analysis was employed to describe the relationship among the studied traits. The first two PCAs accounted for most of the variance, about 78.21% (65.45% and 12.76% by PC1 and PC2, respectively). Therefore, the two PCs were utilized to construct the PC-biplot (Figure 3). A robust positive association was confirmed between grain yield and each of the following: PH, NRPE, NKPR, and HKW. Moreover, a robust positive relationship was observed between grain yield and resistance to both PLB and PSB. On the contrary, days to 50% silking displayed a negative relationship with PLB and PSB.

## 3. Discussion

The pink stem borer, *S. cretica*, the purple-lined borer, *C. agamemnon* and the European corn borer *Ostrinia nubilalis*, are considered the most destructive insects of maize in the Mediterranean region [33]. Therefore, breeding maize hybrids to improve resistance to corn borers alongside high-yielding is a decisive goal to reduce yield losses and ensure food security. Genetic variation plays an important role in developing potential hybrids for improving resistance to corn borers and productivity. In the current study, substantial variations were observed among the evaluated hybrids for corn borer resistance and agronomic traits. These findings indicate the presence of considerable genetic variation among the hybrids that allows for the selection of desired hybrids. In this perspective, Oluwaseun et al. [34], Elmyhun et al. [35], Amegbor et al. [36], and Ajala et al. [37] observed significant genetic variability for various agronomic characteristics in maize. Likewise, Samayoa et al. [38] and Ismail et al. [39] revealed high genetic variability for resistance to different corn borers in maize hybrids.

A selection of superior maize hybrids targets multiple traits, including borer resistance alongside grain yield and its components to increase productivity and adaptability to biotic stresses. A prime objective of this study was to distinguish resistant hybrids to PSB and PLB alongside high-yielding performance. The results indicated that the four crosses, IL1×IL6, IL3×IL7, IL4×IL5, and IL6×IL7, exhibited high resistance to PSB and PLB with high agronomic performance. The identified hybrids displayed a grain yield improvement of 14–29% over the commercial hybrid. This suggests the high ability of these hybrids to compete with commercial checks and would appear as promising candidates for commercial exploitation after further testing. Moreover, these hybrids could be considered potential genotypes that could be used in developing new superior inbred lines [40].

The inheritance of desirable characteristics is crucial for breeding programs. Therefore, exploring the inheritance pattern for resistance to PSB and PLB and agronomic traits is important for selection and breeding. In the current study, both additive and non-additive gene actions were equally crucial in the inheritance of all studied traits, as demonstrated by substantial GCA and SCA effects for all studied traits. On the other hand, the GCA/SCA ratio was less than the unity for agronomic traits, implying that non-additive gene action mainly regulated the inheritance of these characteristics. Accordingly, the crossing is highly effective in enhancing these traits to exploit the heterosis effect. These results are consistent with those of Badu-Apraku et al. [27], Kamara et al. [41], and Makumbi et al. [28], who found that the non-additive gene action was more important for agronomic traits in maize. Contrary to the findings of the present study Badu-Apraku et al. [42], Badu-Apraku et al. [43], Annor et al. [44] and Oyetunde et al. [45] manifested that additive gene action has a predominant role in the inheritance of maize grain yield. Otherwise, the GCA/SCA ratio surpassed the unity for the resistance to PSB and PLB, suggesting that additive gene action mainly modulated the inheritance of these characteristics; hence, recurrent selection could be a feasible method for improving these characteristics. These findings are in harmony with the results of Ajala et al. [30] and Olayiwola et al. [19]. They signified the predominance of additive gene action for stem borer resistance. The predominance of GCA×Y interaction over SCA×Y for most evaluated traits indicated that the additive effects were more impacted by the environment than the non-additive ones. This finding is corroborated by Ismail et al. [39] and Kamara et al. [46].

Breeding superior hybrids with high resistance to borer and agronomic performance are principally determined by the selection of proper parental inbred. The significant and negative GCA effects recorded by the IL1 and IL4 for DTS revealed that these inbred could be potential genetic resources for enhancing earliness. Early flowering saves irrigation water and protects against corn borer attacks [46]. The inbred line IL1 was recognized as a good combiner for reducing plant height which is important for enhancing lodging tolerance. Furthermore, enhancing agronomic performance could be achieved by exploiting IL6 and IL7, which expressed highly significant and positive GCA effects. Additionally, IL5, IL6, and IL7 were recognized as excellent combiners for resistance to PSB and PLB. Consequently, employing these inbred in maize breeding programs could participate in generating progenies with improved borer resistance. Remarkably, the IL6 and IL7 had favorable GCA effects for grain yield as well as being superior combiners for resistance to PSB and PLB. Therefore, these inbred could inherit these valuable alleles from their offspring to generate resistant and high-yielding hybrids. This result confirmed the finding of Olayiwola et al. [19] using a different set of maize inbred lines.

SCA estimates are used to identify desirable newly developed hybrids. In this study, most of the evaluated hybrids had desirable and substantial SCA effects for at least one trait. The result revealed that crosses, IL1×IL4, IL1×IL6, IL2×IL5, IL2×IL7, and IL3×IL6 were identified as having negative SCA effects DTS, which is suitable for improving earliness in maize. Moreover, the crosses IL1×IL6, IL2×IL5, IL3×IL7, IL4×IL5, and IL6×IL7 were recorded to be the best specific combiner for developing high-yielding hybrids. The high SCA effect occurred as a consequence of non-additive gene effects. Out of these crosses, four hybrids, IL1×IL6, IL3×IL7, IL4×IL5, and IL6×IL7, had desirable SCA coupled with high grain yield. The relationship between intended SCA effects and high agronomic performance was also demonstrated by Elmyhun et al. [35] and Kamara et al. [46]. Furthermore, the hybrids IL1×IL5, IL1×IL6, IL3×IL6 and IL3×IL7 appeared as the promising specific combiners for enhancing resistance to the two targeted corn borers in this study (PSB and PLB). Most of these crosses were produced from a good×poor general combiner. In this manner, Kamara et al. [46] elucidated that the presence of at least one good combiner is needed for generating good specific combinations. The hybrid combinations which had superior SCA effects for borer resistance and grain yield are greatly desirable in maize breeding programs. Curiously, the hybrids IL1×IL6, IL3×IL6, and IL3×IL7 possessed high SCA for PSB and PLB resistance and consistently significant advantageous SCA for agronomic performance. Subsequently, these crosses could be efficiently exploited in maize breeding programs to improve resistance to PSB and PLB attacks and maize agronomic performance.

Information on relationships between grain yield and other studied traits could reinforce the efficiency of maize breeding programs [31,47]. The PC-biplot displays an appropriate approach to exploring the associations between the evaluated traits [20,46,48,49]. The results obtained in this study indicate that the improvement of one trait will straightly result in a significant change in other correlated traits owing to their robust association. Strong positive relationships were identified between grain yield and each 100-kernel weight, number of kernels/row, number of rows/ear, and plant height. This implies their significance as beneficial traits for indirect selection for improving grain yield. These results concur with previously published reports that have demonstrated the positive association between maize grain yield and its related traits [20,36,46,50]. Additionally, grain yield was highly correlated with PSB and PLB resistance. This implies that the selection of highly resistant hybrids could contribute to an increase in grain yield. In contrast, the resistance against PSB and PLB attacks was negatively associated with days to 50% silking. Hence, earliness would be favorable for resistance to corn borers. Corresponding results were disclosed by López-Malvar et al. [33] and Ordas et al. [51].

## 4. Materials and Methods

### 4.1. Plant Materials

Seven diverse white maize inbred lines with contrasting levels of resistance to borer attack collected from separate sources were used in the present study. The source and pedigree of the used inbred lines are provided in Appendix A. In the summer season of 2019, the seeds of the seven inbred lines were split and planted on three sowing dates (15, 22, and 29 May) to avoid variations in flowering time and to obtain sufficient hybrid seeds. The inbred lines were planted in 6-m-long rows and 0.70-m between rows. In rows, two seeds were planted per hill and spaced 0.25-m apart. After full emergence and before the first irrigation, maize seedlings were thinned to one per hill. The half-diallel mating design (without reciprocals) was utilized to develop all possible cross combinations among seven inbred lines. Accordingly, 21 F_1_ hybrids were developed.

### 4.2. Field Trials

The obtained twenty-one F_1_ hybrids alongside the high-yielding commercial check hybrid and resistant to PSB and PLB (SC-132) were assessed under natural field infestation with the two borer species (PSB and PLB) in a private farm, Kafr El-Sheikh, Egypt (31°6′ N, 30°56′ E) during two growing seasons of 2020 and 2021. The experimental site is characterized by high temperatures during the maize season, as displayed in Figure 4. In both seasons, the chemical and physical properties of the experimental site were analyzed and described in Appendix A. With three replications, a Randomized Complete Block Design (RCBD) was applied at each growing season. The plots comprised two rows 6-m long and 0.7-m wide. Nitrogen, phosphorus, and potassium fertilizers were used at rates of 286 kg N/ha, 75 kg P_2_O_5_/ha, and 115 kg/K_2_O. The other standard agronomic practices were performed as recommended, except pest control which was entirely avoided.

### 4.3. Data Collection

Days to 50% silking (DTS), plant height (PH), ear height (EH), ear diameter (ED), ear length (EL), number of kernels per row (NKPR), number of rows per ear (NRPE), 100- kernel weight (HKW), and grain yield (GY) for the assessed hybrids were recorded. The DTS was determined when half of the plants in each plot started forming silks. The PH was recorded in cm as the distance from the soil surface to the top of the first tassel branch. The EH was recorded in cm as the distance from the soil surface to the base of the topmost ear. At harvest, ten random ears were collected from each plot to determine ED, EL, NKPR, NRPE, and HKW. Plots were hand-harvested, and grain yield was recorded by utilizing shelled grain weight (adjusted to 15.5% grain moisture content). Grain yield per plot was converted to tons/ha. A hand-held moisture meter was used to determine grain moisture at harvesting time.

Infestation levels of PSB and PLB were monitored weekly from 20 June to 14 September 2020 in the first season and from 25 June to 20 September 2021 in the second growing season. Field surveys were carried out in the morning time. Ten plants were randomly collected using cross diameter technique from each plot. The whole plants were examined in the field, and the stems of selected plants were dissected to monitor the internal tunnels. Different borer species were recorded and utilized as an assessment of damage made by *S. cretica*, in the early stages of plant life and *C. agamemnon*, in the late stages of plant life. The dissected plants with at least one larva or more were considered infected with PSB and/or PLB.

The resistance index was recorded to compare the resistance level of the evaluated hybrids to PSB or PLB. The plants that sheltered larvae of *S. cretica* and/or *C. agamemnon* or showed symptoms of infestation with one of these insects were also dissected to check the presence of larvae. The resistance index to PLB was calculated by using the following equation:$$\mathrm{Resistance}\;\mathrm{to}\;\mathrm{PLB}\;\left(\%\right)=\left(1-\frac{\mathrm{No}.\;\mathrm{PLB}\;\mathrm{infested}\;\mathrm{plants}\;\mathrm{in}\;\mathrm{each}\;\mathrm{plot}}{\mathrm{No}.\;\mathrm{total}\;\mathrm{plants}\;\mathrm{in}\;\mathrm{each}\;\mathrm{plot}}\right)\times 100$$

While the resistance index to PSB was computed by the following equation:$$\mathrm{Resistance}\;\mathrm{to}\;\mathrm{PSB}\;\left(\%\right)=\left(1-\frac{\mathrm{No}.\;\mathrm{PSB}\;\mathrm{infested}\;\mathrm{plants}\;\mathrm{in}\;\mathrm{each}\;\mathrm{plot}}{\mathrm{No}.\;\mathrm{total}\;\mathrm{plants}\;\mathrm{in}\;\mathrm{each}\;\mathrm{plot}}\right)\times 100$$

### 4.4. Statistical Analysis

The analysis of variance was applied for all recorded data using R statistical software version 4.1.1. A combined analysis was performed across the two years where the homogeneity test was non-significant. Means for each trait were estimated based on combined data across years. The angular transformation was employed for the resistance index in accordance with Snedecor and Cochran [52]. To distinguish the significance of variations between means, the least significant difference (LSD) values were calculated. Combining ability analysis was applied according to Griffing’s (1956) method 4 model 1 [23]. A principal component analysis (PCA) was performed utilizing averages of the studied traits to explore their relationships.

## 5. Conclusions

The present study demonstrated substantial genetic variations among tested hybrids for all evaluated traits. There was a predominance of non-additive effects for grain yield and its contributing traits. The inheritance of PSB and PLB resistance was governed by the additive gene effects. The inbred line IL1 was detected to be a good combiner for earliness and developing short-stature genotypes. Additionally, IL6 and IL7 were recognized as excellent combiners to enhance resistance to PSB and PLB and grain yield. The hybrids IL1×IL6, IL3×IL6, and IL3×IL7 combined substantial SCA with high agronomic performance. These hybrids will be further assessed for potential release and commercial cultivation. Robust positive associations were determined among grain yield, its related traits, and resistance to PSB and PLB. This implies their significance as beneficial traits for indirect selection for improving grain yield. Otherwise, the resistance against PSB and PLB was negatively associated with the silking date, indicating that earliness would be favorable for escaping from the borer’s attack. The information obtained from this study could be utilized for improving grain yield and borer resistance in maize breeding programs.

## Figures and Tables

**Figure 1 plants-12-01105-f001:**
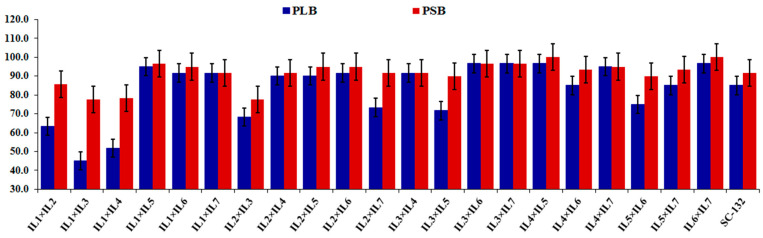
Resistance of the developed 21 F_1_ hybrids and the commercial check (SC-132) for purple-lined borer (PLB) and pink stem borer (PSB) resistance.

**Figure 2 plants-12-01105-f002:**
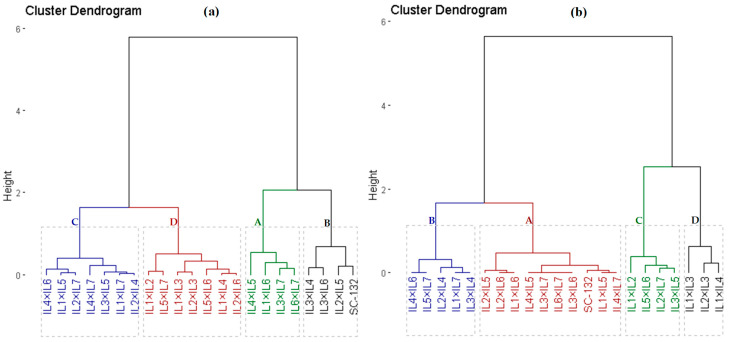
Dendrogram of the phenotypic distances among 21 F_1_ maize hybrids based on the evaluated agronomic traits (**a**) and corn borers resistance measurements (**b**).

**Figure 3 plants-12-01105-f003:**
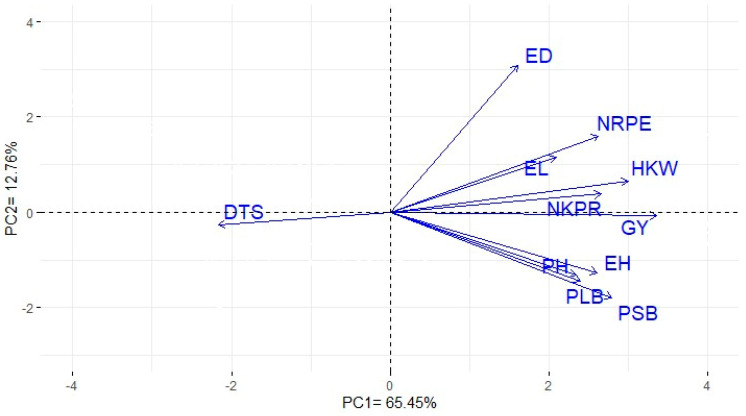
Biplot of the first two principal components for the evaluated agronomic traits and resistance to corn borers of the evaluated hybrids. DTS: days to 50% silking; PH: plant height; EH: ear height; EL: ear length; ED: ear diameter; NRPE: number of rows per ear; NKPR: number of kernels per row; HKW: hundred kernel weight; GY: grain yield (ton ha ^−1^); PLB: purple-lined borer; and PSB: pink stem borer resistance.

**Figure 4 plants-12-01105-f004:**
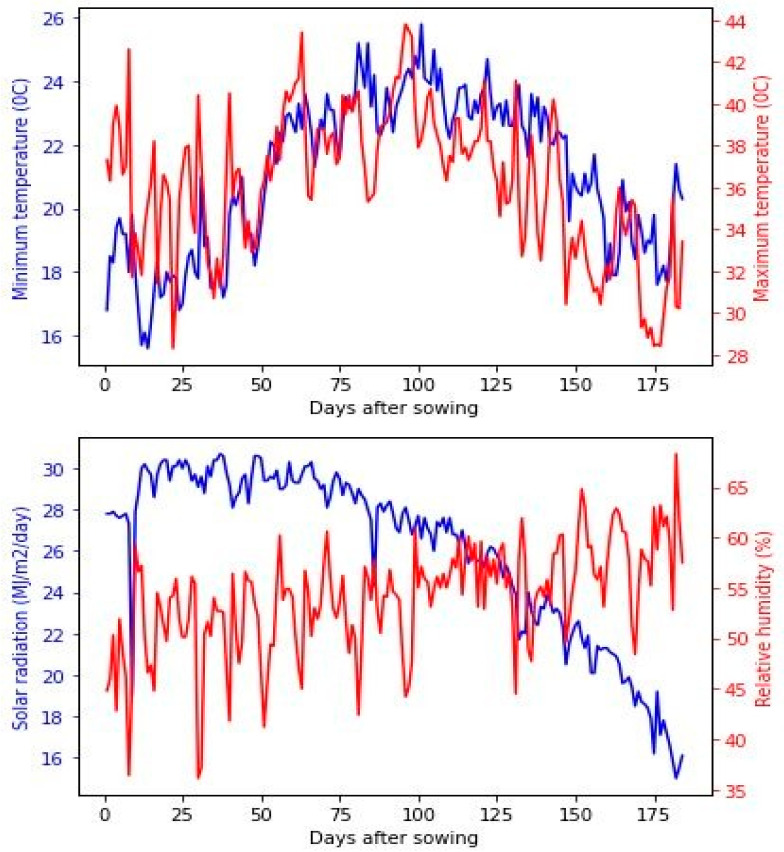
Daily minimum temperature, maximum temperature, and solar radiation for the two growing seasons.

**Table 1 plants-12-01105-t001:** Combined analysis of variance for agronomic and corn borer resistance traits over two seasons.

Source of Variance	DF	Agronomic Traits									Resistance to Corn Borers (%)	
		DTS	PH	EH	EL	ED	NRPE	NKPR	HKW	GY	PLB	PSB
Years (Y)	1	155.76	2369 *	2855 *	65.63	4.08	2.96	85.96	96.55	32.67	50.20	38.20
Replication (Y)	4	22.03	202.0	160.6	8.63	0.84	0.55	15.13	18.31	8.71	38.89	26.97
Hybrids (H)	20	49.21 **	1397 **	583.0 **	9.50 **	0.75 **	4.90 **	47.55 **	36.40 **	13.07 **	1426.51 **	264.7 **
GCA	6	19.98 **	415.4 **	175.2 **	3.43 **	0.21 **	0.85 **	16.40 **	8.79 **	2.48 **	553.28 **	144.0 **
SCA	14	14.87 **	487.6 **	202.5 **	3.05 **	0.27 **	1.97 **	15.61 **	13.56 **	5.16 **	442.17 **	64.33 **
H×Y	20	19.42 **	768.5 **	494.4 **	11.11 **	0.48 **	2.09 **	26.29 **	12.38 **	2.50 **	85.56 *	101.2 **
GCA×Y	6	7.07 **	236.7 **	193.1 **	4.81 **	0.18 **	0.80 **	11.30 **	4.03 **	0.87 **	32.96 *	35.45 **
SCA×Y	14	6.22 **	264.5 **	152.7 **	3.23 **	0.15 **	0.65 **	7.68 **	4.17 **	0.82 **	26.61 *	33.00 **
Error	80	1.46	134.1	31.28	0.56	0.16	0.53	5.48	1.66	0.40	43.53	13.97
GCA/SCA		1.34	0.85	0.87	1.12	0.81	0.43	1.05	0.65	0.48	1.25	2.24
GCA×Y/SCA×Y		1.14	0.89	1.26	1.48	1.20	1.23	1.47	0.97	1.06	1.24	1.07

* and ** indicate *p*-value < 0.05 and 0.01, respectively; DF: degree of freedom; DTS: days to 50% silking; PH: plant height; EH: ear height; EL: ear length; ED: ear diameter; NRPE: the number of rows/ear; NKPR: the number of kernels/row; HKW: hundred kernel weight; GY: grain yield (ton ha^−1^); PLB: purple-lined borer; and PSB: pink stem borer resistance.

**Table 2 plants-12-01105-t002:** Agronomic performance of the developed 21 hybrids and the commercial check (SC-132) over two growing seasons.

Hybrid	DTS	PH	EH	EL	ED	NRPE	NKPR	HKW	GY
IL_1_×IL_2_	65.33	156.67	92.50	16.17	4.53	14.00	28.33	29.53	3.84
IL_1_×IL_3_	66.17	140.83	77.00	14.33	4.55	13.00	27.17	28.18	3.32
IL_1_×IL_4_	59.17	170.83	87.50	14.60	4.28	13.33	25.83	30.03	3.50
IL_1_×IL_5_	63.67	146.67	77.50	13.47	4.15	12.83	31.67	28.92	4.34
IL_1_×IL_6_	62.00	167.50	98.33	17.18	5.04	14.33	32.67	34.08	7.03
IL_1_×IL_7_	61.67	143.33	79.17	14.63	4.28	13.17	23.83	29.65	4.57
IL_2_×IL_3_	66.50	137.67	70.00	15.33	4.98	12.83	29.17	26.33	3.22
IL_2_×IL_4_	65.67	169.17	84.17	14.83	4.50	13.33	24.83	30.07	4.62
IL_2_×IL_5_	62.00	145.00	86.67	14.80	4.38	13.83	30.83	31.08	6.44
IL_2_×IL_6_	68.17	165.00	94.17	16.72	4.43	13.17	31.17	31.73	3.56
IL_2_×IL_7_	62.17	162.50	88.33	13.45	4.28	14.50	28.83	32.65	4.41
IL_3_×IL_4_	62.17	169.17	94.17	16.50	4.28	12.33	31.50	27.63	5.45
IL_3_×IL_5_	65.17	161.67	90.83	13.47	4.18	14.00	32.17	31.52	4.50
IL_3_×IL_6_	61.67	146.67	90.00	16.15	4.78	14.17	32.83	32.20	5.71
IL_3_×IL_7_	62.17	182.50	97.50	15.75	4.60	14.67	30.67	33.02	7.29
IL_4_×IL_5_	65.17	194.50	108.33	17.05	4.70	15.00	33.35	33.88	7.94
IL_4_×IL_6_	61.33	168.33	96.83	14.80	3.57	11.17	27.17	25.07	4.20
IL_4_×IL_7_	61.33	140.83	68.67	14.30	4.38	13.00	29.67	30.03	4.83
IL_5_×IL_6_	70.17	164.83	92.50	13.25	3.69	13.00	27.67	30.87	3.70
IL_5_×IL_7_	67.00	161.67	90.50	13.38	4.28	13.00	27.83	29.28	3.97
IL_6_×IL_7_	68.67	180.00	98.33	14.83	4.62	14.50	33.33	34.17	7.52
SC-132	63.17	157.17	89.17	15.60	4.47	12.83	29.83	31.08	6.13
LSD_0.05_	1.33	13.30	6.42	0.86	0.45	0.83	2.69	1.48	0.73
LSD_0.01_	1.77	17.63	8.52	1.14	0.60	1.10	3.57	1.96	0.97

DTS: days to 50% silking; PH: plant height; EH: ear height; EL: ear length; ED: ear diameter; NRPE: number of rows/ear; NKPR: number of kernels/row; HKW: hundred kernel weight; and GY: grain yield (ton ha^−1^).

**Table 3 plants-12-01105-t003:** General combining ability effects for the assessed seven inbred lines for all studied traits over two years.

Inbred Lines	DTS	PH	EH	EL	ED	NRPE	NKPR	HKW	GY	Resistance to Corn Borers (%)	
										PLB	PSB
IL1	−1.39 **	−7.71 **	−4.06 **	0.08	0.07	−0.05	−1.56 **	−0.49	−0.62 **	−11.86 **	−4.86 **
IL2	0.98 **	−5.68 *	−3.29 *	0.26	0.14	0.15	−0.82	−0.29	−0.72 **	−4.19 **	−2.52 **
IL3	−0.22	−5.18	−2.56	0.31	0.19 *	0.02	1.24 *	−0.79 *	−0.04	−5.52 **	−3.86 **
IL4	−2.02 **	9.69 **	1.48	0.42 *	−0.14	−0.55 **	−0.99	−1.22 **	0.17	2.48	0.14
IL5	1.64 **	1.99	2.81 *	−0.92 **	−0.21 *	0.15	1.24 *	0.54	0.24	3.14 *	3.16 **
IL6	1.41 **	5.59 *	7.58 **	0.59 **	−0.06	−0.11	1.51 **	1.06 **	0.40 **	7.81 **	4.14 **
IL7	−0.39	1.29	−1.96	−0.73 **	0.001	0.39 *	−0.62	1.19 **	0.58 **	8.14 **	3.81 **
LSD (gi)_0.05_	0.55	5.50	2.66	0.35	0.19	0.34	1.11	0.61	0.30	3.14	1.78
LSD (gi)_0.01_	0.73	7.30	3.53	0.47	0.25	0.46	1.48	0.81	0.40	4.16	2.36

* and ** indicate *p*-value < 0.05 and 0.01, respectively, DTS: days to 50% silking; PH: plant height, EH: ear height; EL: ear length; ED: ear diameter; NRPE: number of rows/ear; NKPR: number of kernels/row; HKW: hundred kernel weight; GY: grain yield (ton ha^−1^); PLB: purple-lined borer; and PSB: pink stem borer resistance.

**Table 4 plants-12-01105-t004:** Specific combining ability effects of 21 test-crosses for all studied traits over two years.

Hybrid	DTS	PH	EH	EL	ED	NRPE	NKPR	HKW	GY	Resistance to Corn (%) Borers	
										PLB	PSB
IL1×IL2	1.59 **	9.32	11.13 **	0.83 *	−0.09	0.41	1.17	−0.16	0.24	−3.56	1.67
IL1×IL3	3.62 **	−7.01	−5.10	−1.05 **	−0.13	−0.46	−2.07	−1.01	−0.97 **	−20.56 **	−5.33 **
IL1×IL4	−1.58 **	8.12	1.37	−0.89 *	−0.06	0.44	−1.17	1.27 *	−1.00 **	−21.89 **	−8.50 **
IL1×IL5	−0.74	−8.34	−9.97 **	−0.69	−0.13	−0.76 *	2.43 *	−1.61 **	−0.23	20.78 **	6.83 **
IL1×IL6	−2.18 **	8.89	6.10 *	1.52 **	0.61 **	1.01 **	3.17 **	3.04 **	2.30 **	12.78 **	4.17 *
IL1×IL7	−0.71	−10.98 *	−3.53	0.29	−0.21	−0.66	−3.53 **	−1.53 *	−0.34	12.44 **	1.17
IL2×IL3	1.59 **	−12.21 *	−12.87 **	−0.23	0.25	−0.82 *	−0.80	−3.06 **	−0.96 **	−4.89	−7.67 **
IL2×IL4	2.56 **	4.42	−2.73	−0.84 *	0.10	0.24	−2.90 *	1.11	0.22	8.78 *	2.50
IL2×IL5	−4.78 **	−12.04 *	−1.57	0.46	0.05	0.04	0.87	0.36	1.97 **	8.11*	2.83
IL2×IL6	1.62 **	4.36	1.17	0.87 *	−0.05	−0.36	0.93	0.49	−1.07 **	5.11	1.83
IL2×IL7	−2.58 **	6.16	4.87	−1.08 **	−0.26	0.48	0.73	1.27 *	−0.40	−13.56 **	−1.17
IL3×IL4	0.26	3.92	6.53 *	0.78 *	−0.17	−0.62	1.70	−0.82	0.38	11.78 **	3.83 *
IL3×IL5	−0.41	4.12	1.87	−0.92 *	−0.21	0.34	0.13	1.29 *	−0.65 *	−8.89 *	−0.83
IL3×IL6	−3.68 **	−14.48 **	−3.73	0.26	0.25	0.78 *	0.53	1.46 *	0.40	11.44 **	4.83 **
IL3×IL7	−1.38 *	25.66 **	13.30 **	1.17 **	0.00	0.78 *	0.50	2.14 **	1.81 **	11.11 **	5.17 **
IL4×IL5	1.39 *	22.09 **	15.33 **	2.55 **	0.65 **	1.91 **	3.53 **	4.09 **	2.58 **	8.11*	5.17 **
IL4×IL6	−2.21 **	−7.68	−0.93	−1.20 **	−0.64 **	−1.66 **	−2.90 *	−5.24 **	−1.32 **	−8.22 *	−2.50
IL4×IL7	−0.41	−30.88 **	−19.57 **	−0.39	0.12	−0.32	1.73	−0.41	−0.87 **	1.44	−0.50
IL5×IL6	2.96 **	−3.48	−6.60 *	−1.42 **	−0.44 *	−0.52	−4.63 **	−1.20	−1.89 **	−18.89 **	−8.83 **
IL5×IL7	1.59 **	−2.34	0.93	0.03	0.08	−1.02 **	−2.33 *	−2.92 **	−1.80 **	−9.22 **	−5.17 **
IL6×IL7	3.49 **	12.39 *	4.00	−0.02	0.27	0.74 *	2.90 *	1.45 *	1.59 **	−2.22	0.50
LSD Sij_0.05_	1.09	10.86	5.24	0.70	0.37	0.68	2.20	1.21	0.60	6.19	3.50
LSD Sij_0.01_	1.44	14.40	6.96	0.93	0.49	0.90	2.91	1.60	0.79	8.20	4.65

* and ** indicate *p*-value < 0.05 and 0.01, respectively. DTS: days to 50% silking; PH: plant height; EH: ear height; EL: ear length; ED: ear diameter; NRPE: number of rows per ear; NKPR: number of kernels per row; HKW: hundred kernel weight; GY: grain yield (ton ha^−1^); PLB: purple-lined borer; and PSB: pink stem borer resistance.

## Data Availability

The data presented in this study are available upon request from the corresponding author.

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
