# Peer review of "Combining Ability and Inheritance Nature of Agronomic Traits and Resistance to Pink Stem (Sesamia cretica) and Purple-Lined (Chilo agamemnon) Borers in Maize"

_plants, 2023, doi:10.3390/plants12051105_

Round 1
Reviewer 1 Report
This paper addresses conventional plant breeding for resistance to maize stem borer pests. The study was conducted over 2 seasons and under natural infestation. Using natural infestation is not ideal when resistance evaluations are done under field conditions, due to the differences in agronomic traits of maize lines, and the preference that female moths have for plants of different ages.
Please use scientific names in title.
Indicate the families of the borer species when you mention them for the 1st time.
The text font in the supplementary files differs within tables. Please correct this.
Line 18. Is Ostrinia not also of high importance in the Mediterranean region?? Rephrase.
Line 19. Rephrase unclear. The frequent use of chemical pesticides has resulted din the evolution of resistance to various………………
Line 54. rephrase
Line 57. Replace the word severely with ‘’that’’.
Line 61. This statement about dead heart is incorrect. It is not leaf feeding that causes dead heart but feeding deep inside the whorl and death of the growth point.
Line 67. Reference number 12 is a very old reference to cite in this context.
Line 71. Replace overrun with ‘’pest outbreaks’’
Line 432. Rephrase. Infestation levels of PSB and PLB were monitored….
Line 439. Do you mean one larva per plant? Or per plot? Is this not too low to get accurate data?
Lines 441-444. Please rephrase the text to indicate that a resistance index was calculated. It is not resistance that was measured, but an Index was used to compare resistance levels between entries. Also line 457. It is incorrect to refer to resistance percentages. There cannot be such a thing.
Line 447. Please also explain if plants that were infested by PLB, was included in the equation of PSB %, and vica versa.
No information / description is provided about the temperature recordings in the Methods section.
Lines 307-309. Unclear what you mean??
The reference list needs attention. Many minor mistakes.
Author Response
Dear Editor,
We would like to thank you and the reviewers for the time and efforts devoted to our manuscript entitled “Combining Ability and Inheritance Nature of Agronomic Traits and Resistance to Purple-Lined and Pink Stem Borers in Maize” (plants-2231920). We have revised the manuscript according to the comments and suggestions pointed out by the reviewers. We have addressed the comments of the reviewers in a point-by-point below in red color; in addition, we have highlighted all the associated changes made to the manuscript using track changes.
Yours sincerely,
Authors
Responses to Reviewers' Comments
Reviewer 1:
This paper addresses conventional plant breeding for resistance to maize stem borer pests. The study was conducted over 2 seasons and under natural infestation. Using natural infestation is not ideal when resistance evaluations are done under field conditions, due to the differences in agronomic traits of maize lines, and the preference that female moths have for plants of different ages.
Re: We would like to thank the Reviewer for providing constructive criticism to improve the quality of the manuscript. The field trials of this study were performed within a region of maize production and the infestation of pink stem and purple-lined appears and is common naturally in the region. The study aimed at assessing the preference of female insects and their natural attraction to the evaluated hybrids due to their characteristics and secreted substances to attract or repel insects without researcher intervention.
Please use scientific names in the title.
Re: Done as requested
Indicate the families of the borer species when you mention them for the 1st time.
Re: The families of borer species have been added as requested, please see lines 18 and 19 in the revised version
The text font in the supplementary files differs within tables. Please correct this.
Re: The font in the supplementary file has been corrected
Line 18. Is Ostrinia not also of high importance in the Mediterranean region?? Rephrase.
Re: Yes you are right, the sentence has been modified as suggested (line 19)
Line 19. Rephrase unclear. The frequent use of chemical pesticides has resulted in the evolution of resistance to various………………
Re: The sentence has been rephrased as suggested (lines 21-22)
Line 54. rephrase
Re: The sentence has been rephrased (lines 55-57)
Line 57. Replace the word severely with ‘’that’’.
Re: “severely” has been replaced by ‘’that’’ (line 60)
Line 61. This statement about the dead heart is incorrect. It is not leaf feeding that causes dead heart but feeding deep inside the whorl and death of the growth point.
Re: The statement has been corrected as requested (lines 63-65)
Line 67. Reference number 12 is a very old reference to cite in this context.
Re: Reference number 12 as well as all references in the text have been updated as suggested.
Line 71. Replace overrun with ‘’pest outbreaks’’
Re: Done as suggested (line 76)
Line 432. Rephrase. Infestation levels of PSB and PLB were monitored….
Re: Rephrased as suggested (line 394)
Line 439. Do you mean one larva per plant? Or per plot? Is this not too low to get accurate data?
Re: Ten plants randomly were collected for survey using a cross-diameter technique from each plot. The whole plants were examined in the field and the stems of selected plants were dissected to monitor the internal tunnels. Different borer species were recorded and utilized as an assessment of damage made by S. cretica, in the early stages of plant life and C. agamemnon in the late stages of plant life. The dissected plants with at least one larva or more were considered infected with PSB and/or PLB. More clarification has been added (lines 397-402).
Lines 441-444. Please rephrase the text to indicate that a resistance index was calculated. It is not resistance that was measured, but an Index was used to compare resistance levels between entries. Also line 457. It is incorrect to refer to resistance percentages. There cannot be such a thing.
Re: The sentence has been rephrased as suggested (lines 405-409 and 422)
Line 447. Please also explain if plants that were infested by PLB were included in the equation of PSB %, and vice versa.
Re: When the two species were found inside the same plant, each specie was calculated separately in its own equation. A separate equation was provided for each specie (lines 413-417)
No information/description is provided about the temperature recordings in the Methods section.
Re: More information has been added to the text (lines 369-370) and the figure caption (Figure 4, lines 378-380)
Lines 307-309. Unclear what you mean??
Re: The paragraph has been rephrased (253-260)
The reference list needs attention. Many minor mistakes.
Re: The reference list has been carefully revised and corrected
Thanks so much for your accurate review that contributed considerably to improve our manuscript.
Reviewer 2 Report
Dear all, the article is interesting and contains useful information regarding maize breeding. My suggestions are marked in yellow in the attached text and explained in the comments.
At the most I suggest checking and changing the older quotes which bring statements that may no longer be true due to the changes that eventually occur over the years. Whenever possible, search for more current citations.

Author Response
Responses to Reviewers' Comments
Reviewer 2:
Dear all, the article is interesting and contains useful information regarding maize breeding. My suggestions are marked in yellow in the attached text and explained in the comments.
At the most, I suggest checking and changing the older quotes which bring statements that may no longer be true due to the changes that eventually occur over the years. Whenever possible, search for more current citations.
Re: We would like to thank the Reviewer for his time dedicated to our manuscript and his positive assessment of our work.
The comments and suggestions pointed out by reviewer 2 in the PDF file are addressed below:
Line 26: Why did you use this hybrid, is it resistant? Mention the characteristic that led to its use.
Re: The used check hybrid (SC-132) is a high-yielding commercial hybrid resistant to pink-stem borer and purple-lined borer. More details have been added to the text (please see lines 29-30 and 364-365 in the revised version)
Line 57: Are these problems still occurring to this day without any improvement? I ask this because the references cited here are very old. The most recent one is 21 years old and the oldest 50 years old. Aren't there more recent reports about what was said in this sentence?
Re: The paragraph has been revised and old references have been replaced by Soujanya et al. 2021 (Sci. Rep. 11, 14770) and Ismail et al. 2019 (Indian J. of Agric. Sci. 89, 1953-1958.) please see line 61.
Line 60: I believe you can use a more recent reference here as well.
Re: The reference has been replaced by Horgan et al. 2021 (Crop Prot. 142, 105513) please see line 65.
Line 65: I suggest reviewing all the literature cited in that paragraph. We are stating many things here that have already passed more than 10 years and we don't know if they continue like this until today.
Re: All references in the manuscript have been revised and updated to be eight were published in 2023, ten in 2022, eight in 2021, four in 2020, three in 2019, two in 2015, two in 2014, one in 2011, one in 2010 and two in 2008.
Line 74-76 This statement needs literary citation.
Re: Reference has been added (Olayiwola et al. 2021 Euphytica, 217, 14) line 81
Line 83 and 84????
Re: The sentence has been revised and modified (please see lines 88-90)
Line 149: Is this commercial check hybrid a PLB and PSB resistant or susceptible hybrid? This is important information for me to know about the actual superiority of the tested hybrids.
Re: Yes, the used check hybrid (SC-132) is a high-yielding commercial hybrid resistant to PLB and PSB. More information have been added to the text (lines 29-30 and 364-365)
Line 168: It would be interesting to carry out this analysis including the commercial check hybrid. In this way, we could verify whether it would be allocated together with the resistant or susceptible hybrids.
Re: The Dendogram has been reconstructed including the commercial check hybrid as suggested (Figure 2 a and b).
Line 170: I suggest using capital letters, bold, or color to better differentiate groups during reading
Re: The letters have been capitalized and bolded as suggested (lines 162-170)
Line 293-295 This quote is 29 years old, I don't know if we can still claim that.
Re: The paragraph has been revised and the references have been updated. Moreover, all references in the manuscript have been revised and updated to be eight were published in 2023, ten in 2022, eight in 2021, four in 2020, three in 2019, two in 2015, two in 2014, one in 2011, one in 2010 and two in 2008.
Thanks so much for your review which contributed considerably to improve our manuscript.
Round 2
Reviewer 1 Report
The authors did a good job of revising the manuscript. The methods section was improved by providing much more detail.
Line 67. Rephrase. It is not he damage that is caused by tunnels.
Line 244. Delete the PSB and PLD. These acronyms were already provided.
Line 403. Replace ‘’infected’’ with ‘’infested’’. Fungi infects = insects infest.
Author Response
Dear Editor,
We would like to thank you and the reviewers for the time and efforts devoted to our manuscript ” (plants-2231920). We have addressed the new comments of Reviewer 1.
Yours sincerely,
Authors
Responses to Reviewer 1:
Comments and Suggestions for Authors
The authors did a good job of revising the manuscript. The methods section was improved by providing much more detail.
Re: We would like to thank Reviewer 1 for his time dedicated to our manuscript and his positive assessment of our revision.
Line 67. Rephrase. It is not the damage that is caused by tunnels.
Re: The sentence has been rephrased to be "Besides, the tunnels are caused by larvae entering the stem weaken the plant and consequently increase stalk lodging and reduce grain yield." (line 67)
Line 244. Delete the PSB and PLD. These acronyms were already provided.
Re: The PSB and PLD have been deleted as requested (line 245)
Line 403. Replace ‘’infected’’ with ‘’infested’’. Fungi infect = insects infest.
Re: “infected’’ has been replaced with ‘’infested’’ as suggested (line 404)